# PGR and Its Application Method Affect Number and Length of Runners Produced in 'Maehyang' and 'Sulhyang' Strawberries

**Chen Liu [1], Ziwei Guo [1], Yoo Gyeong Park [2], Hao Wei [1] and Byoung Ryong Jeong [1,2,3,*]**

1   Department of Horticulture, Division of Applied Life Science (BK21 Plus Program), Graduate School of Gyeongsang National University, Jinju 52828, Korea; chenliu215@gmail.com (C.L.); guoziwei1230@gmail.com (Z.G.); oahiew@gmail.com (H.W.)
2   Institute of Agriculture and Life Science, Gyeongsang National University, Jinju 52828, Korea; ygpark615@gmail.com
3   Research Institute of Life Science, Gyeongsang National University, Jinju 52828, Korea
*   Correspondence: brjeong@gmail.com; Tel.: +82-010-6751-5489

**Abstract:** Vegetative propagation using runner plants is an important method to expand the cultivation area for the strawberry (*Fragaria × ananassa* Duch.). However, excessively long runners need an increased total amount of nutrients and energy to receive elongation from mother plants, which may lead to poor growth or reduced output. The use of plant growth regulators (PGRs) is an adoptable way to solve such problems. The objectives of this experiment were to study the effects of PGRs and their application methods on the growth and development of runners, runner plants, and mother plants, and also to find effective ways to control the number and length of runners without harmful side effects. Chlormequat chloride (CCC), 6-benzylaminopurine (BA), and ethephon (ETH) at a concentration of 100 mg·L$^{-1}$ were applied via three different methods: injection into crowns, medium drench, and foliar spray. The results showed that BA injection into crowns was the most effective combination among all treatments, which prominently shortened the length of runners and increased the number of runners and leaves on a single plant. Furthermore, plants with BA solution injection tended to produce stronger runners with higher fresh and dry weights, without affecting the health states of mother plants. The ETH solution seemed to have toxic effects on plants, by leading many dead leaves and weak runners, and increased activities of antioxidant enzymes. Other than the injection method, the other two application methods of the CCC solution did not significantly affect the growth and development of both cultivars. Runner plants grown for 30 days were not affected by any treatments, and they were in similar conditions. Overall, BA injection into crowns is recommended for controlling the number and length of strawberry runners.

**Keywords:** chlormequat chloride; ethephon; *Fragaria × ananassa*; 6-benzylaminopurine; runner

## 1. Introduction

The strawberry (*Fragaria × ananassa*), a herbaceous perennial crop species from the Rosaceae family, is one of the most popular fruit crops with great economic values. As a berry, the strawberry is full of vitamins and minerals that are good for human health [1]. It is commercially grown in approximately 80 countries [2]. In 2013, global strawberry production exceeded 7.7 million tons [3]. Runners, or stolons, are stems that grow on the ground surface, with several nodes that are capable of generating adventitious roots and daughter plants because of their meristematic tissues during the growth and development stages [4]. Adventitious roots are available at the second, fourth, or sixth nodes, where the newly-formed plants can be used for propagation. However, in commercial

production, runners and runner plants are generally removed by growers [5] because mother plants do not stop sending out runners and runners keep growing until some actions are taken, which may result in the deterioration of the mother plants' condition. Runners and runner plants need a large amount of productive energy and nutrition from mother plants, causing mother plants to have reduced outputs. It is also reported that asexual reproduction reduces fruit yields [6].

Although overgrowth of runners may influence the health of mother plants, as mentioned above, sometimes growers deliberately keep runners on mother plants to produce new generation plants. In the Republic of Korea, especially in the southern areas of the country, strawberry growers usually control environmental conditions to force strawberries to bloom as early as possible, usually by late October to early November, in the cultivation season. From November to the following February is the season to produce/harvest fruits. March to April is a suitable time for growing runners, while May and June are the best months to harvest runner plants to be used as new mother plants. Runner plants are often collected and sold or planted to expand the planting area. July to September is the time when the new generation plants grow. Such a strawberry production pattern forms a complete cycle in the Republic of Korea and is repeated year after year. Using runner plants for propagation is a method with easy operation, high propagation coefficient, and low cost. Cutting propagation by runner plants is an important way for strawberry reproduction because strawberry runner plants are easily produced and rooted [7], and the new plants retain their parents' good traits. To sum up, controlling the length and number of runners to limit harmful effects to their mother plants and offspring is important. Unfortunately, few studies have been carried out concerning this issue.

According to former research carried out by Kumar et al., potato runners can be controlled by phytohormones like indoleacetic acid (IAA) and gibberellins (GA) [8]. Plant growth regulator (PGR) is an artificial chemical phytohormone analogue that is essential for regulating plant growth and development in agriculture, and can be applied to control plant size, flowering, fruiting, and output [9]. It is also reported that PGRs, such as chlormequat chloride (commercially available under the trade name cycocel, CCC) and ethephon (ETH), can shorten certain parts of a plant [10]. Due to differences in properties and reactions of different plant tissues to PGRs, effects of PGRs may vary greatly. Even with the same PGR, different application methods may lead to different results. There are many methods of applying PGRs, such as spraying, drenching, and dipping. Therefore, it is worth trying because taking different approaches may yield unexpected benefits.

Discovered by Professor Tolbert at Michigan State University in the 1950s, CCC is the first plant growth regulator used on plants [11]. It is a synthetic PGR that is antagonistic to gibberellins (GA), while GAs are phytohormones that regulate plants' developmental processes, such as stem elongation, dormancy, and germination [12]. The CCC inhibits cell elongation without inhibiting cell division. Studies have shown that CCC can decrease the growth of stems, leaves, and runners of potato [13] and thicken the stem of mung beans [14] effectively by controlling vein growth and lodging. The CCC application results in dwarfed plants; thickened stalks [11]; darkened, greened, and thickened leaves; increased chlorophyll content; and a well-developed root system. Nowadays, CCC is widely used in agricultural production to slow down the stem growth, while enhancing flowering in many crops.

The ETH was discovered in 1965 and was first registered as a pesticide in the U.S. in 1973 [15]. It has low toxicity, and in the U.S. it is registered for use on ornamental plants as well as wheat, barley, apple, blackberry, cherry, grape, pineapple, cucumber, tomato, pepper, coffee, cotton, and tobacco [16]. The ETH is antagonistic to IAA, and is easily converted into ethylene in aqueous solutions of pH 5 or higher [17]. Ethylene interferes the growth processes of plants, and is a potent regulator of plant growth and ripeness. According to previous studies, exogenous ETH on turfgrass reduces the mowing frequency [18], which means it prevents or slows down the growth processes of turfgrass.

6-benzylaminopurine (BA) is a first generation man-made cytokinin that plays an important role in plant cell division, fruit growth acceleration, shoot formation, fruit setting, and yield increases [19]. Furthermore, BA increases plants' stress resistance [20] and therefore is widely used in horticulture and agriculture [21].

As discussed above, three PGRs have been widely researched, developed, and used commercially for the past half century to manipulate plant shape, form, and overall crop quality in agriculture and horticulture. The hypothesis of this study comes from the fact that some PGRs can control the length of plant organs, such as potato runners [13]. Especially, it is hypothesized that BA induces a large number of cells to divide, which would lead to more runners emerging and competing with each other, resulting in the decreased length of the runners. The objectives of this study were to realize the effects of PGRs and their application methods on the growth and development of strawberry runners, runner plants, and mother plants, and also to find out methods to control the number and length of runners without the harmful effects of PGRs on the plants.

## 2. Materials and Methods

### 2.1. Plant Materials and Culture Conditions

The strawberry cultivars used were 'Maehyang' and 'Sulhyang'. Plants were purchased from a strawberry farm (Sugok-myeon, Jinju, Gyeongsangnam-do, Republic of Korea) and maintained in the BVB Medium (Bas Van Buuren Substrate, EN-12580, De Lier, The Netherlands). The experiment was carried out in a glasshouse at Gyeongsang National University in the Republic of Korea. The culture environment had 23/17 °C day/night average temperatures, 70–80% relative humidity, and a natural photoperiod of 14 h or so. Plastic pots (Green-100, Danong Co., Namyangju, Republic of Korea) were used as growing containers, and plants were treated after confirming all plants produced at least one runner and the length of runner was approximately 5 cm. To count the number of new leaves and runners, the redundant leaves and runners were removed before the first treatment, leaving three leaves and one runner on each plant. Treatments were given weekly on Friday mornings for one month (1–31 May 2018).

When treatments were finished, 10 healthy and uniform runner plants in each treatment generated on the second node of runners were selected and stuck into the BVB Medium contained in 21-cell zigzag trays (21-Zigpot/21 cell tray, Daeseung, Jeonju, Republic of Korea). A fogging system (UH-303, JB Natural Co. Ltd., Gunpo, Republic of Korea) was used to promote induction of roots for about 9 days, and plants were checked to find out if there were any toxic effects from PGR treatments. The cultivation environment of runner plants had 32/21 °C day/night temperatures (average), 75–85% relative humidity, and a natural photoperiod of approximately 14.5 h. This stage lasted for another 30 days from June 1 to 30, 2018.

For maintenance, a greenhouse multipurpose nutrient solution (in $mg \cdot L^{-1}$ $Ca(NO_3)_2 \cdot 4H_2O$ 737.0, $KNO_3$ 343.4, $KH_2PO_4$ 163.2, $K_2SO_4$ 43.5, $MgSO_4 \cdot H_2O$ 246.0, $NH_4NO_3$ 80.0, Fe-EDTA 15.0, $H_3BO_3$ 1.40, $NaMoO_4 \cdot 2H_2O$ 0.12, $MnSO_4 \cdot 4H_2O$ 2.10, and $ZnSO_4 \cdot 7H_2O$ 0.44 (electrical conductivity 0.8 $dS \cdot m^{-1}$)) was provided by drenching the growing medium daily.

### 2.2. Plant Growth Regulators Tested

CCC, BA, and ETH (MB-C4219, MB-B5812, and MB-E5360, respectively. MB Cell, Seoul, Republic of Korea) were used in this experiment. Unfortunately, there was little research done on the application of these PGRs and the proper concentration of these PGRs for strawberry. In accordance with agricultural production practices and guidelines for the use of PGRs, we found that growers usually treat cucumber, tomato, eggplant, and melon with CCC solution at 100–500 $mg \cdot L^{-1}$; rose and chrysanthemum with BA solution at 50–200 $mg \cdot L^{-1}$; and cucumber, melon, and watermelon at 100–500 $mg \cdot L^{-1}$. In order to prevent plant damage from highly concentrated PGR solutions and also to ensure the consistency of the experimental concentration, 100 $mg \cdot L^{-1}$ for all three PGRs was used in this experiment. The CCC and ETH were dissolved in distilled water, while BA was dissolved in a 1N NaOH solution. After testing, it was determined that application of a 5 mL of PGR for each treatment was the most appropriate for each plant, as 5 mL of the solution to a plant was the most appropriate,

and 5 mL is sufficient to make the medium permeated and cover all the leaves of a plant without much loss.

### 2.3. Methods of Supplying PGR Solution

CCC, BA, and ETH at a concentration of 100 mg·L$^{-1}$ were applied using three different mathods: injection into crowns, medium drench, and foliar spray.

#### 2.3.1. Injection

To inject the PGR solution, crowns were pierced with an injection syringe (12 mL, Jung Rim Medical Industrial Co. Ltd., Seoul, Republic of Korea), and 5 mL solution was injected into the plant each time.

#### 2.3.2. Drench

An injection syringe (25 mL, Jung Rim Medical Industrial Co. Ltd., Seoul, Republic of Korea) without a needle was used to drench the growing medium with 5 mL PGR solution.

#### 2.3.3. Spray

For spraying, a sprayer was filled with the PGR solution at the total calculated volume according to the number of plants so that 5 mL is foliage-sprayed evenly per plant.

### 2.4. Measurements of Growth and Morphological Parameters

After 30 days of treatment, on 1 June 2018, growth parameters such as length of the longest runner, average length between the crown and the second node, runner diameter, number of runners per plant, fresh and dry weights of runners, and number of new leaves of the mother plants were measured. It is noteworthy that when the length of the longest runner was measured, most runners only had four nodes and very few of them had five nodes. To make the data more comparable, only length of four nodes was measured. After another 30 days, on 1 July 2018, growth parameters such as average length of shoot and root, crown diameter, fresh and dry weights of shoot and root, number of leaves, leaf length, leaf width and thickness, and chlorophyll level, of runner plants were measured.

Dry weights of shoot, root, and whole plant were measured after 72 h of drying in a drying oven (FO-450M, Jeio Technology Co. Ltd., Daejeon, Republic of Korea) at 70 °C. Diameters of the runner, crown, and leaf thickness were measured using a Vernier caliper (CD-20CPX, Mitutoyo Korea Co., Gunpo, Republic of Korea) at the widest points. The chlorophyll level was measured with a chlorophyll meter (SPAD-502, Konica Minolta Inc., Japan) on three healthy leaves in each plant to be averaged.

Samples for physiological analysis were taken from leaves of three randomly selected plants in each treatment among young and healthy leaves with uniform sizes and same conditions. Samples were fixed with liquid nitrogen as quickly as possible.

### 2.5. Measurements of Contents of Starch, Soluble Sugar, and Protein

#### 2.5.1. Soluble Sugar and Starch

The contents of soluble sugar and starch were assayed according to the Anthrone colorimetric method [22]. For each treatment, a 0.2 g leaf sample was ground into the homogenate with distilled water and then transferred into a 15 mL centrifugal tube. The volume was adjusted to 6 mL and extracted in boiling water for 30 min. Then, the residue was filtered for starch extraction and the remaining solution was adjusted to 15 mL. A 0.1 mL of the extracted sample solution was added to 3 new 15 mL centrifugal tubes for each treatment, and 0.1 mL distilled water was used as the control. Then, 1.9 mL distilled water, 0.5 mL 2% Anthrone ethylacetate, and 5 mL 98% H$_2$SO$_4$ were added in that order into the tubes. All tubes were submerged in boiling water for 10 min. The absorbance was

measured at 630 nm with a spectrophotometer (Uvikon 992, Kotron Instrumentals, Milano, Italy) after cooling the samples to room temperature.

The residue of sugars was used to assay the starch content. The residues from different treatments were added to tubes with 5 mL distilled water and boiled for 15 min to extract starch. Afterwards, 0.7 mL of 9.2 mol·L$^{-1}$ perchloric acid was added to each tube, and tubes were placed in boiling water for additional 15 min. The volumes were adjusted to 15 mL after cooling down. The contents from the tubes were filtered, and the same volume of extracted samples, distilled water, Anthrone ethylacetate, and $H_2SO_4$ were added into new tubes to measure the absorbance at 485 nm. The soluble sugar and starch contents were calculated according to the prepared standard curves.

### 2.5.2. Protein

Total protein content was measured based on the reaction of Coomassie brilliant blue G-250 with proteins by measuring the absorbance at 595 nm with a spectrophotometer according to the method of Bradford [23]. The $Na_2HPO_4$ and $NaH_2PO_4$ were mixed in distilled water according to protocol for the phosphate buffer. Afterwards, 0.058 g EDTA-Na$_2$, 0.1 mL 0.05% Triton X solution, and 4.0 g 2% PVP were added to the phosphate buffer and the pH was adjusted to 7.0 to finish the working buffer. A 0.1 g of the leaf sample and 1.5 mL of working buffer were taken and ground into the homogenate in an ice box. This mixture was centrifuged at 13,000 rpm, 4 °C for 20 min with a centrifuge (5430 R, Eppendorf AG, Hamburg, Germany), and then the supernatant was transferred to new e-tubes. For measurement, 50 μL of the supernatant was mixed with 1,450 μL of Bradford's reagent, and was held still for 5–10 min. A standard curve was made by using Bovine serum albumin.

### *2.6. Measurements of Antioxidant Enzymes Activities*

### 2.6.1. Superoxidase Dismutase (SOD)

According to the protocol of Beauchamp and Fridovich [24], the SOD activity was assayed by measuring the capacity to inhibit the photochemical reduction of nitroblue tetrazolium (NBT). The measurement was conducted with a 3 mL reaction mixture containing 50 mM phosphate buffer (pH 7.8), 14.5 mM methionine, 2.25 μM NBT, 60 μM riboflavin, 30 μM EDTA, and 0.1 mL of the enzyme extract. This reaction solution was incubated for 20 min under fluorescent lamps at an illuminance of 4000 lux. A tube containing the enzyme was kept in dark and served as the blank, while the control tube without enzyme extracts was kept in light. The absorbance was taken at 560 nm, and calculations were made by using an extinction coefficient of 100 mM$^{-1}$·cm$^{-1}$.

### 2.6.2. Peroxidase (POD)

According to the protocol described by Sadasivam and Manickam [25], a 0.2 M phosphate buffer (pH 6.0), 0.076 mL guaiacol solution, 0.1 mL enzyme extract, and 0.112 mL 30% hydrogen peroxide solution were prepared for the enzyme assay. An increase in the absorbance was recorded at 470 nm. The time was recorded in 30-s intervals until the decrease became constant. The extinction coefficient was 6.39 per micromole.

### 2.6.3. Catalase (CAT)

The total CAT activity was measured by the method of Aebi [26]. The assay system consisted of a 0.15 M phosphate buffer (pH 7.0), 0.31 mL 30% hydrogen peroxide solution, and 0.1 mL of the enzyme extract in the final volume of 3 mL. The decrease in the absorbance was recorded at 240 nm. The molar extinction coefficient of $H_2O_2$ at 240 nm was 0.004 μmol$^{-1}$·cm$^{-1}$.

*2.7. Statistical Analysis*

The data were analyzed with SAS (SAS 9.4, SAS Institute Inc., Cary, NC, USA). The experimental results were subjected to an analysis of variance (ANOVA) and Duncan's multiple range tests. OriginPro 9.0 (OriginLab Co., Northampton, MA, USA) was used for graphing.

## 3. Results

*3.1. Effects of PGR and Application Method on Runners and Mother Plants*

As shown in Figure 1 and Table 1, significant differences were observed between the two cultivars. Runners of strawberry 'Sulhyang' were much longer and stronger than those of 'Maehyang', and PGR affected growth and development of runners in terms of their length; diameter; and, especially, runner number. The application methods also had significant effects on growth and development of runners, such as number, length, diameter, and fresh weight. As for the effect of PGR in combination with application method, diverse effects on number, length, diameter, and dry weight of the runners were found. For example, plants treated with PGRs by injection method shortened length of runners.

For strawberry 'Maehyang', all three PGRs shortened runner length when injected. The treatment of BA injected into crowns induced the greatest number of runners and leaves per plant (Figure 1B). All treatments increased runner diameter, and it was most pronounced in the treatment of BA injected into crowns. The greatest fresh weight of runners was obtained in the treatments of ETH drench, BA injection, and BA drench among all treatments. The ETH solution at a concentration of 100 mg·L$^{-1}$ may have had toxic or senescing effects on the mother plants, as its injection and drench caused many leaves to die (Figure 1C). Furthermore, ETH drench tended to prevent the formation of runners. For strawberry 'Sulhyang', all treatments except the BA injection and BA drench tended to prevent runner induction. The BA injection resulted in the greatest number of runners per plant (Figure 1B). The injection of either ETH or BA was effective in controlling length of runners. Fresh and dry weights of plants in all treatments were not significantly different from those in the control group. There was little difference in number of new leaves and runner diameter among treatments. Drenching medium with a 100 mg·L$^{-1}$ ETH solution may have had toxic or senescing effects on the strawberry plants as it resulted in many dead leaves in this cultivar also (Figure 1C). In summary, injection of a 100 mg·L$^{-1}$ BA solution into crowns of mother plants of both 'Maehyang' and 'Sulhyang' strawberry was effective in controlling number and length of the runners.

**Table 1.** Effect of plat growth regulators (PGR) and application method on growth and development of runners of 'Maehyang' and 'Sulhyang' strawberry measured at 30 days after treatment initiation.

| Cultivar (C) | PGR (P) | Treatment Method (M) | Runner | | | | | |
|---|---|---|---|---|---|---|---|---|
| | | | Number | Length of the Longest Runner (cm) | Length of the 1st & 2nd Internode (cm) | Diameter (mm) | Fresh Weight (g) | Dry Weight (g) |
| 'Maehyang' | CCC | Injection | 1.7 ± 0.2 b [z] | 74.1 ± 3.1 b | 30.9 ± 2.0 b | 2.40 ± 0.04 cd | 5.17 ± 0.35 bc | 0.27 ± 0.02 a |
| | | Drench | 1.5 ± 0.2 b | 85.0 ± 5.8 ab | 37.6 ± 4.0 ab | 2.54 ± 0.10 bc | 6.36 ± 0.52 a–c | 0.32 ± 0.05 a |
| | | Spray | 2.0 ± 0.1 b | 87.8 ± 2.4 a | 38.5 ± 1.6 ab | 2.42 ± 0.12 cd | 5.56 ± 0.57 a–c | 0.29 ± 0.07 a |
| | BA | Injection | 3.5 ± 0.4 a | 74.9 ± 2.7 b | 32.1 ± 1.5 b | 2.75 ± 0.24 a | 6.80 ± 0.86 ab | 0.39 ± 0.04 a |
| | | Drench | 1.7 ± 0.2 b | 83.4 ± 2.1 ab | 37.8 ± 1.3 ab | 2.49 ± 0.06 bc | 7.23 ± 0.83 a–c | 0.35 ± 0.06 a |
| | | Spray | 1.8 ± 0.2 b | 86.9 ± 4.0 a | 40.0 ± 1.7 a | 2.43 ± 0.08 cd | 5.48 ± 0.30 a–c | 0.25 ± 0.04 a |
| | ETH | Injection | 1.7 ± 0.2 b | 73.9 ± 2.1 b | 31.3 ± 1.9 ab | 2.41 ± 0.07 cd | 5.29 ± 0.44 bc | 0.26 ± 0.01 a |
| | | Drench | 1.3 ± 0.2 b | 83.5 ± 6.5 ab | 33.8 ± 3.1 ab | 2.70 ± 0.08 ab | 7.26 ± 0.80 a | 0.32 ± 0.05 a |
| | | Spray | 1.5 ± 0.2 b | 81.8 ± 3.6 ab | 36.3 ± 1.3 ab | 2.34 ± 0.10 cd | 4.83 ± 0.57 c | 0.27 ± 0.05 a |
| 'Sulhyang' | CCC | Injection | 2.0 ± 0.4 bc | 72.8 ± 1.5 a–c | 28.1 ± 1.8 ab | 2.11 ± 0.13 c | 6.10 ± 0.62 a–c | 0.28 ± 0.02 bc |
| | | Drench | 2.2 ± 0.3 bc | 81.3 ± 3.5 a | 31.1 ± 2.8 ab | 2.24 ± 0.09 a–c | 3.27 ± 0.96 c | 0.29 ± 0.02 a–c |
| | | Spray | 1.7 ± 0.3 bc | 77.5 ± 4.8 ab | 30.8 ± 2.2 ab | 2.32 ± 0.09 a–c | 7.06 ± 0.71 a | 0.35 ± 0.03 ab |
| | BA | Injection | 3.5 ± 0.2 a | 64.2 ± 4.5 c | 27.9 ± 1.3 ab | 2.40 ± 0.10 a–c | 3.45 ± 0.15 a–c | 0.37 ± 0.03 a |
| | | Drench | 2.2 ± 0.2 bc | 72.6 ± 2.2 a–c | 34.0 ± 2.7 a | 2.23 ± 0.09 a–c | 5.41 ± 0.78 a–c | 0.23 ± 0.04 c |
| | | Spray | 2.3 ± 0.4 b | 79.8 ± 2.6 a | 33.4 ± 1.2 a | 2.18 ± 0.08 bc | 4.53 ± 0.54 c | 0.27 ± 0.04 bc |
| | ETH | Injection | 1.3 ± 0.2 c | 66.1 ± 3.6 c | 25.5 ± 2.4 b | 2.22 ± 0.13 a–c | 4.94 ± 0.71 a–c | 0.23 ± 0.03 c |
| | | Drench | 2.0 ± 0.3 bc | 67.9 ± 3.5 bc | 28.8 ± 2.0 ab | 2.65 ± 0.13 a | 6.69 ± 0.85 ab | 0.27 ± 0.04 bc |
| | | Spray | 1.8 ± 0.3 bc | 68.1 ± 2.9 bc | 30.7 ± 1.3 ab | 2.60 ± 0.09 ab | 6.12 ± 0.60 a–c | 0.31 ± 0.03 a–c |
| F-test [y] | | C | * | *** | *** | ** | NS | NS |
| | | P | *** | * | NS | * | NS | NS |
| | | M | ** | *** | *** | NS | ** | NS |
| | | C × P | NS | NS | NS | * | *** | NS |
| | | C × M | NS | NS | NS | NS | NS | NS |
| | | P × M | *** | * | NS | * | NS | * |
| | | C × P × M | NS | NS | NS | NS | NS | NS |

z Mean separation within columns for each cultivar by Duncan's multiple range test at *p* < 0.05. y NS, *, **, and ***: nonsignificant or significant at *p* ≤ 0.05, 0.01, and 0.001, respectively.

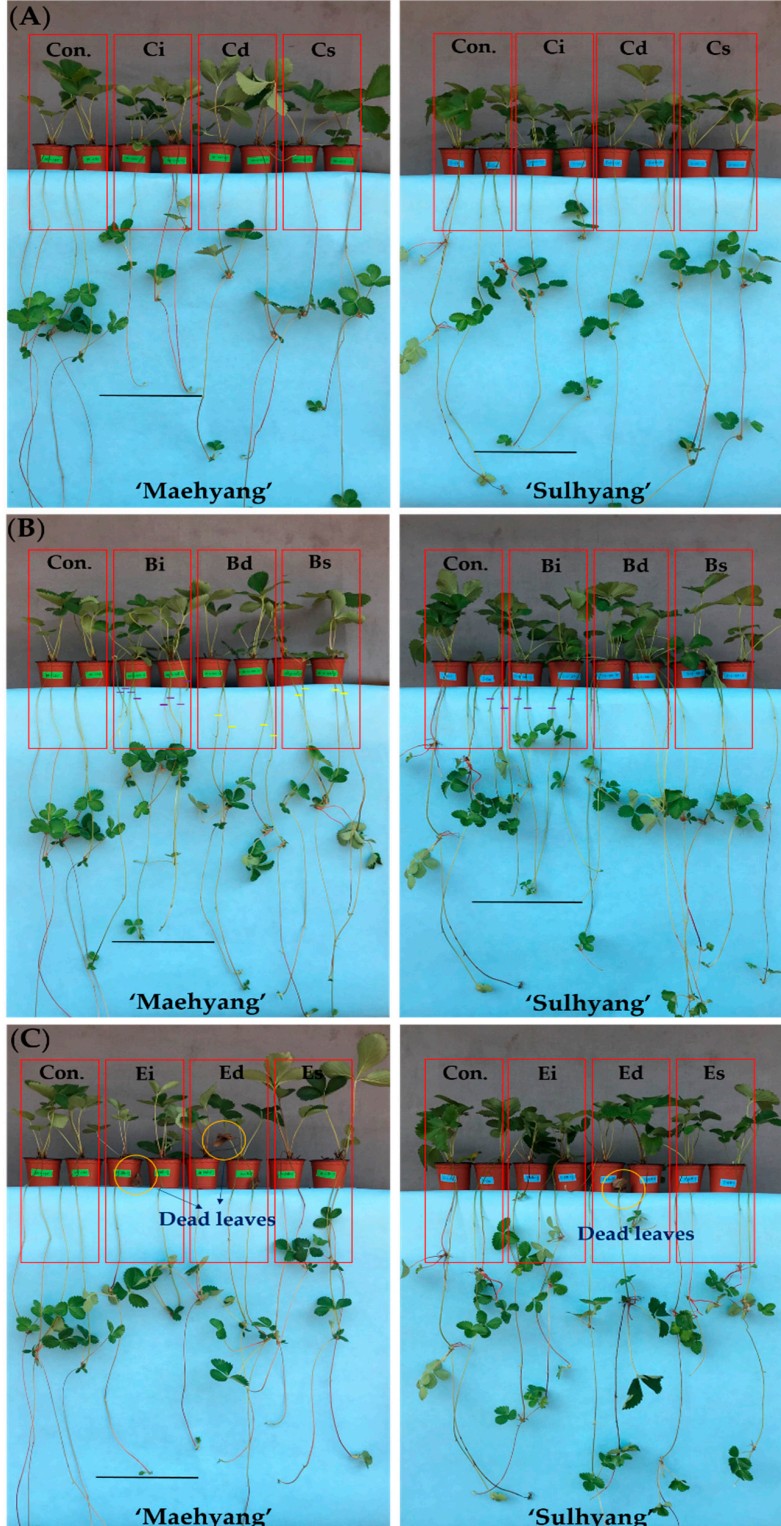

**Figure 1.** The effects of (**A**) chlormequat chloride (CCC), (**B**) 6-benzylaminopurine (BA), and (**C**) ethephon (ETH) at a concentration of 100 mg·L$^{-1}$ and the application method (injection, drench, and spray) on the number and length of runners of strawberries 'Maehyang' and 'Sulhyang' observed 30 days after treatment initiation in strawberries 'Maehyang' and 'Sulhyang': The letters C, B, and E stand for CCC, BA, and ETH, respectively; I, d, and s stand for injection into crowns, medium drench, and foliar spray, and Con. is the control.

## 3.2. The Effects of the PGR Solution and the Application Method on Endogenous Compounds

An analysis of the endogenous compounds showed that the starch, soluble sugar, and protein contents were differently affected by the treatments (Figure 2). The BA injection, BA foliar spray, and ETH injection resulted in lower contents of starch in strawberry 'Maehyang'. All CCC applications, regardless of the application method and BA injection, led to similarly lower starch contents in 'Sulhyang'. For 'Maehyang', CCC injection and all BA treatments resulted in the greatest soluble sugar contents, while BA drench, ETH drench, and ETH spray led to the greatest soluble sugar contents in 'Sulhyang'. The greatest protein contents were obtained in the CCC injection, BA injection, BA spray, and ETH injection for 'Maehyang' and in CCC injection, BA injection, and BA drench for 'Sulhyang'.

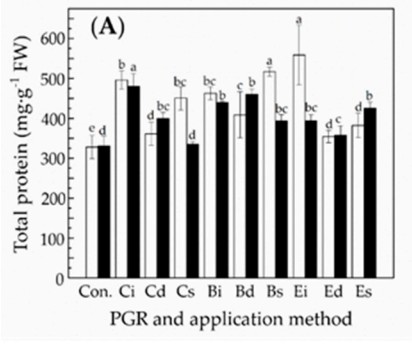
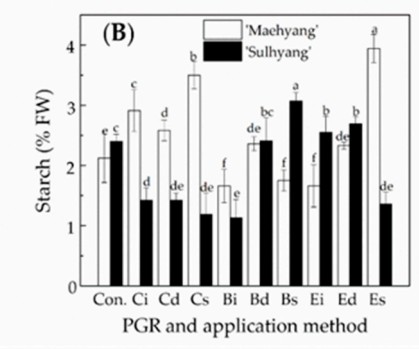

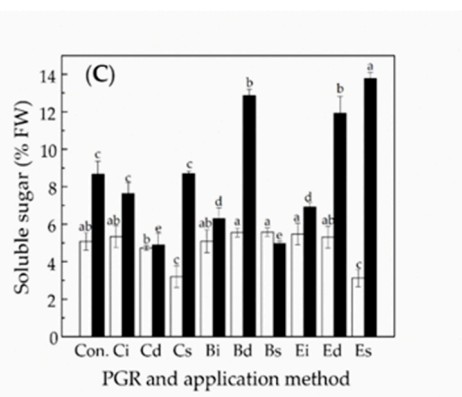

**Figure 2.** The effects of chlormequat chloride (CCC), 6-benzylaminopurine (BA), and ethephon (ETH) at a concentration of 100 mg·L$^{-1}$ and their application method (injection, drench, and spray) on the content of (**A**) total protein, (**B**) starch, and (**C**) soluble sugars in leaves of strawberries 'Maehyang' and 'Sulhyang' sampled 30 days after treatment initiation. Vertical bars indicate the standard error (*n* = 3). Means accompanied by different letters are significantly different (*p* < 0.05) according to the Duncan's multiple range test: The letters C, B, and E stand for CCC, BA, and ETH, respectively; I, d, and s stand for injection into crowns, medium drench, and foliar spray, and Con. is the control.

## 3.3. The Effects of the PGR Solution and the Application Method on the Activities of Antioxidant Enzymes

All treatments resulted in increased SOD activity in 'Maehyang' compared to the control group. The CCC drench, ETH drench, and ETH spray resulted in markedly increased SOD activity compared to CCC injection, CCC spray, and BA injection. The SOD activity in 'Sulhyang' in the CCC spray, ETH drench, and ETH injection treatments was greater than that of the control group. The CCC drench and BA injection resulted in the lowest SOD activity. The CCC injection, CCC spray, and BA spray resulted in a considerably higher POD activity than other treatments in 'Maehyang'. Similarly, foliar spray of either CCC or BA resulted in much higher POD activity than other treatments, while all other treatments did not increase POD activity than the control in 'Sulhyang'. The CAT activity in 'Maehyang' in all treatments increased compared to the control, with the exception of BA injection and CCC drench, which had much higher CAT activity than the control. For 'Sulhyang', BA injection

resulted in the lowest CAT activity, followed by CCC drench and CCC spray, and all other treatments led to high CAT activities (Figure 3).

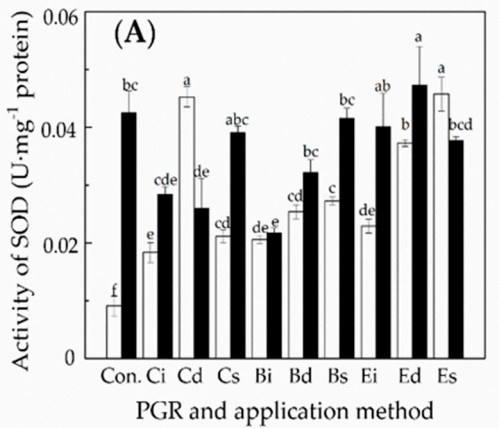
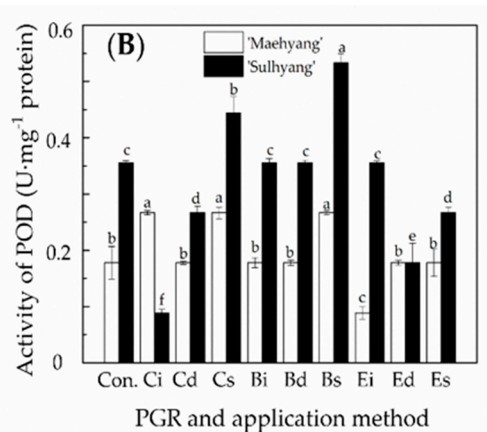

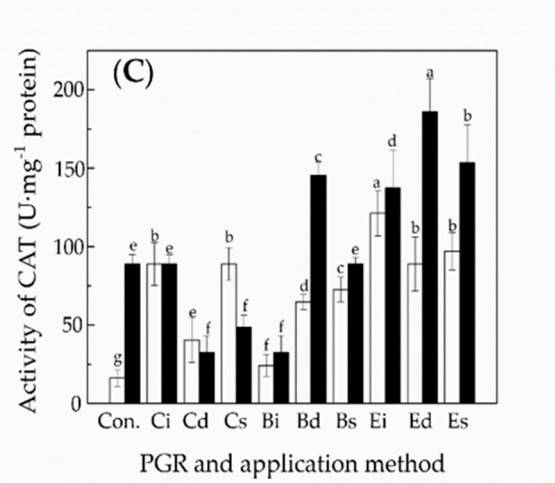

**Figure 3.** The effects of the chlormequat chloride (CCC), 6-benzylaminopurine (BA), and ethephon (ETH) at a concentration of 100 mg·L$^{-1}$ and their application method (injection, drench, and spray) on the activities of (**A**) SOD (superoxidase dismutase), (**B**) POD (peroxidase), and (**C**) CAT (catalase) in the leaves of strawberry 'Maehyang' and 'Sulhyang' sampled 30 days after treatment initiation. Vertical bars indicate the standard error ($n = 3$). Means accompanied by different letters are significantly different ($p < 0.05$) according to the Duncan's multiple range test: The letters C, B, and E stand for CCC, BA, and ETH, respectively; I, d, and s stand for injection into crowns, medium drench, and foliar spray, and Con. is the control.

### 3.4. The Effects of the PGR Solution and the Application Method on the Runner Plants

With respect to the next generation, all runner plants with PGRs applied in different methods were observed to grow better than or at least as well as the control group for both cultivars. Across the two cultivars, there were significant differences that include the growth data of the shoot, root, and leaf, while within the same cultivar there was little difference between plants in different treatments (Figure 4). According to the results of the F-test, the PGRs and the application method exhibited little influence on the runner plants, which means that different PGR treatments in this study had no harmful effects on the next generation of strawberry plants. Data are shown in Tables 2 and 3.

Table 2. Effect of PGR and application method on growth and development of runner plants of strawberries 'Maehyang' and 'Sulhyang' grown for 30 days.

| Cultivar (C) | PGR (P) | Application Method (M) | Shoot | | | Root | |
|---|---|---|---|---|---|---|---|
| | | | Length (cm) | Crown Diameter (mm) | Fresh Weight (g) | Length of Longest Root (cm) | Fresh Weight (g) |
| 'Maehyang' | CCC | Injection | 24.2 ± 1.0 a [z] | 7.20 ± 0.21 a | 6.31 ± 0.32 a–c | 18.2 ± 1.3 | 1.76 ± 0.14 |
| | | Drench | 24.6 ± 1.2 a | 7.13 ± 0.10 a | 7.01 ± 0.24 ab | 17.2 ± 0.7 | 1.84 ± 0.26 |
| | | Spray | 24.4 ± 1.6 a | 7.60 ± 0.37 a | 7.28 ± 0.75 a | 16.0 ± 1.2 | 1.75 ± 0.16 |
| | BA | Injection | 24.9 ± 1.0 a | 7.36 ± 0.53 a | 6.17 ± 0.39 a–c | 18.3 ± 0.1 | 1.78 ± 0.20 |
| | | Drench | 17.9 ± 0.6 d | 7.02 ± 0.13 a | 3.54 ± 0.15 d | 16.9 ± 1.6 | 1.60 ± 0.14 |
| | | Spray | 19.5 ± 2.2 cd | 6.74 ± 0.24 b | 5.01 ± 0.39 c | 16.6 ± 0.3 | 1.86 ± 0.26 |
| | ETH | Injection | 24.0 ± 1.6 a | 7.71 ± 0.36 a | 6.83 ± 0.22 ab | 17.7 ± 1.1 | 1.94 ± 0.16 |
| | | Drench | 22.7 ± 1.8 ab | 7.54 ± 0.41 a | 7.08 ± 0.76 ab | 19.1 ± 1.0 | 2.15 ± 0.27 |
| | | Spray | 24.8 ± 1.0 a | 7.19 ± 0.30 a | 5.93 ± 0.34 a–c | 18.0 ± 0.4 | 1.72 ± 0.21 |
| 'Sulhyang' | CCC | Injection | 26.1 ± 1.3 ab | 6.91 ± 0.13 ab | 5.97 ± 0.92 ab | 15.5 ± 2.0 | 1.70 ± 0.33 |
| | | Drench | 26.1 ± 0.7 ab | 6.61 ± 0.30 ab | 5.46 ± 1.20 ab | 17.8 ± 1.0 | 1.89 ± 0.30 |
| | | Spray | 27.1 ± 1.7 a | 7.20 ± 0.36 a | 6.13 ± 0.70 ab | 17.5 ± 0.6 | 2.11 ± 0.15 |
| | BA | Injection | 23.1 ± 0.1 b–d | 6.82 ± 0.12 ab | 4.21 ± 0.30 b | 15.8 ± 0.8 | 1.93 ± 0.15 |
| | | Drench | 27.3 ± 0.5 a | 7.20 ± 0.39 a | 7.14 ± 0.45 a | 17.6 ± 0.1 | 1.94 ± 0.37 |
| | | Spray | 22.1 ± 0.5 cd | 6.85 ± 0.13 ab | 3.88 ± 0.32 b | 15.8 ± 0.4 | 1.96 ± 0.17 |
| | ETH | Injection | 24.4 ± 0.8 a–c | 6.46 ± 0.24 ab | 4.19 ± 1.79 b | 16.2 ± 0.4 | 1.73 ± 0.30 |
| | | Drench | 24.3 ± 0.4 a–c | 5.43 ± 0.12 c | 5.64 ± 0.47 ab | 17.1 ± 0.6 | 1.91 ± 0.31 |
| | | Spray | 21.1 ± 1.7 d | 6.22 ± 0.33 b | 4.24 ± 0.46 b | 15.3 ± 0.7 | 1.59 ± 0.22 |
| F-test [y] | C | | * | *** | ** | NS | NS |
| | P | | *** | NS | ** | NS | NS |
| | M | | NS | NS | NS | NS | NS |
| | C × P | | * | *** | * | NS | NS |
| | C × M | | * | NS | NS | NS | NS |
| | P × M | | NS | NS | * | NS | NS |
| | C × P × M | | * | NS | * | NS | NS |

z Mean separation within columns for each cultivar by Duncan's multiple range test at p < 0.05. y NS, *, **, and ***: nonsignificant or significant at $p \leq 0.05$, 0.01, and 0.001, respectively.

**Table 3.** Effect of PGR and application method on growth and development of runner plants of strawberries 'Maehyang' and 'Sulhyang' grown for 30 days.

| Cultivar (C) | PGR (P) | Method (M) | The Largest Leaf | | | | | |
|---|---|---|---|---|---|---|---|---|
| | | | Number | Length (cm) | Width (cm) | Thickness (mm) | Petiole Diameter (mm) | Chlorophyll (SPAD) |
| 'Maehyang' | CCC | Injection | 5.0 ± 0.0 a[z] | 7.2 ± 0.2 ab | 4.8 ± 0.1 ab | 0.55 ± 0.05 | 2.43 ± 0.08 ab | 35.7 ± 2.1 b |
| | | Drench | 5.0 ± 0.0 a | 7.5 ± 0.2 ab | 5.4 ± 0.1 a | 0.55 ± 0.04 | 2.44 ± 0.06 ab | 38.8 ± 0.3 ab |
| | | Spray | 5.0 ± 0.0 a | 7.5 ± 0.5 ab | 5.2 ± 0.4 ab | 0.53 ± 0.05 | 2.46 ± 0.12 ab | 38.2 ± 1.2 ab |
| | BA | Injection | 5.0 ± 0.0 a | 7.5 ± 0.5 ab | 5.5 ± 0.5 a | 0.60 ± 0.04 | 2.44 ± 0.10 ab | 39.5 ± 1.5 ab |
| | | Drench | 5.0 ± 0.0 a | 5.2 ± 0.2 d | 3.7 ± 0.1 c | 0.51 ± 0.05 | 2.04 ± 0.14 c | 38.5 ± 0.9 ab |
| | | Spray | 3.0 ± 0.0 c | 5.8 ± 0.2 cd | 4.2 ± 0.1 bc | 0.51 ± 0.04 | 2.28 ± 0.05 bc | 39.3 ± 0.6 ab |
| | ETH | Injection | 4.0 ± 0.0 b | 7.7 ± 0.7 a | 5.3 ± 0.5 ab | 0.54 ± 0.04 | 2.43 ± 0.11 ab | 39.8 ± 0.8 a |
| | | Drench | 4.7 ± 0.3 a | 6.9 ± 0.7 a–c | 4.9 ± 0.6 ab | 0.57 ± 0.04 | 2.66 ± 0.15 a | 38.1 ± 1.5 ab |
| | | Spray | 4.7 ± 0.3 a | 6.8 ± 0.7 a–c | 4.9 ± 0.1 ab | 0.53 ± 0.02 | 2.33 ± 0.07 a–c | 39.6 ± 1.0 ab |
| 'Sulhyang' | CCC | Injection | 5.3 ± 0.3 ab | 8.0 ± 0.2 a–d | 5.6 ± 0.2 ab | 0.34 ± 0.01 | 2.51 ± 0.18 a | 32.6 ± 4.4 b |
| | | Drench | 5.0 ± 0.6 ab | 7.7 ± 0.7 a–d | 5.6 ± 0.6 ab | 0.32 ± 0.01 | 2.22 ± 0.10 a–c | 35.4 ± 1.7 ab |
| | | Spray | 5.7 ± 0.3 a | 8.3 ± 0.5 a–c | 5.8 ± 0.3 ab | 0.33 ± 0.02 | 2.50 ± 0.06 a | 37.2 ± 2.0 ab |
| | BA | Injection | 5.3 ± 0.3 ab | 6.9 ± 0.4 b–d | 4.9 ± 0.2 b | 0.35 ± 0.01 | 2.03 ± 0.01 c | 36.2 ± 2.3 ab |
| | | Drench | 5.3 ± 0.3 ab | 8.7 ± 0.6 a | 6.2 ± 0.4 a | 0.33 ± 0.02 | 2.49 ± 0.09 a | 34.5 ± 0.7 ab |
| | | Spray | 5.3 ± 0.3 ab | 6.7 ± 0.5 cd | 4.9 ± 0.3 b | 0.31 ± 0.03 | 1.96 ± 0.06 c | 35.3 ± 1.5 ab |
| | ETH | Injection | 5.7 ± 0.3 a | 8.3 ± 0.4 ab | 5.6 ± 0.3 ab | 0.34 ± 0.02 | 2.37 ± 0.11 ab | 40.2 ± 1.2 a |
| | | Drench | 5.0 ± 0.6 ab | 7.8 ± 0.3 a–d | 5.2 ± 0.2 ab | 0.33 ± 0.02 | 2.38 ± 0.01 ab | 37.9 ± 0.1 ab |
| | | Spray | 4.3 ± 0.3 b | 6.6 ± 0.6 d | 4.8 ± 0.5 b | 0.31 ± 0.03 | 2.10 ± 0.16 bc | 37.1 ± 2.4 ab |
| F-test[y] | C | | *** | ** | ** | *** | * | ** |
| | P | | * | ** | * | NS | ** | NS |
| | M | | ** | NS | NS | NS | NS | NS |
| | C × P | | ** | NS | NS | NS | NS | NS |
| | C × M | | NS | * | NS | NS | NS | NS |
| | P × M | | NS | NS | NS | NS | NS | NS |
| | C × P × M | | * | * | ** | NS | *** | NS |

z Mean separation within columns for each cultivar by Duncan's multiple range test at *p* < 0.05. y NS, *, **, and ***: nonsignificant or significant at *p* ≤ 0.05, 0.01, and 0.001, respectively.

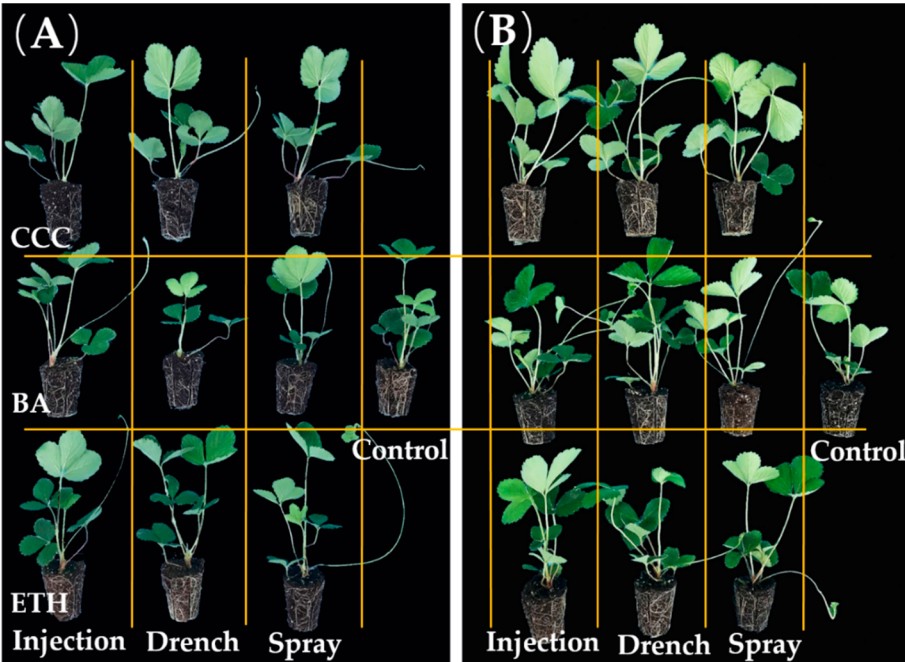

**Figure 4.** The effects of the chlormequat chloride (CCC), 6-benzylaminopurine (BA), and ethephon (ETH) at a concentration of 100 mg·L$^{-1}$ and the application method (injection, drench, and spray) on the morphology of strawberries (**A**) 'Maehyang' and (**B**) 'Sulhyang' runner plants grown for 30 days.

## 4. Discussion and Conclusions

### 4.1. Discussion

Starch is the main form of storage carbohydrate in plants and is important in the carbon economy of many organs, tissues, and cell types of plants [27]. Soluble sugars play an important role in plant growth and developmental processes. They provide energy and mid-metabolites, and also act as signals, regulating the vital movements of plants [28]. The BA was reported to help in accumulating starch in *Lemna minor* [29] and ETH in increasing starch content in apple [30], which are in agreement with results of this study. Medium drenching or foliar spray of BA or ETH led to higher starch content in both cultivars of strawberry plants, while BA injection resulted in the lowest starch content for both cultivars. Root drenching or foliar spray of BA or ETH led to increased starch contents in both cultivars of strawberry plants, while BA injection decreased starch content to the lowest level for both cultivars. On the other hand, BA injection induced an increased level of soluble sugar content in both cultivars. Presumably, some functions of BA help in transforming starch into soluble sugars, and the intensity of this transformation is related to different plant tissues. It was observed that injection of any of three PGRs resulted in shortened runner length for both cultivars, while drenching or spraying PGRs did not. This indicates that PGRs have different mobile characteristics and induced effects depending on the location on the strawberry plants where they are applied, just as hypothesized. The actual mechanisms behind this need to be investigated further.

It is reported that CCC is highly mobile in both xylem and phloem tissues, and is rapidly absorbed and translocated [31]. However, CCC shows different mobility characteristics depending on the species. It was observed that CCC had high mobility in wheat but slow uptake/movement in barley, making CCC more effective in wheat than in barley [32]. In this study, strawberry plants with CCC treatments displayed a similar number of new leaves and runners, length of the first node, percentage of plants with new runners, and runner dry weights, while no significant differences were observed among the plants with respect to the application method. Therefore, it can be concluded that CCC is highly mobile in strawberries too. However, CCC injection into crowns resulted in a more pronounced

shortening of the runner length compared to foliar spray or medium drench. This is probably because CCC injection works directly on the crown where runners are generated, while foliar spray and medium drench result in reduced CCC dosage during the transportation process, making CCC relatively less effective in shortening the runner length.

Truernit et al. (2006) found in *Arabidopsis thaliana* that cytokinin regulates the expression of invertase and transports hexose [33]. As a kind of cytokinin, BA has the effect of inhibiting chlorophyll, nucleic acid, and protein decomposition in the leaves, and various functions such as transporting amino acids, auxin, and mineral salts to plant parts exposed to it [34]. Another important characteristic of BA is its poor mobility in plants; furthermore, its physiological effects are limited to the treatment site and its vicinity. That is the reason why BA injection into crowns induced many more new runners and leaves but medium drench and foliar spray with BA did not. Due to BA helping in accumulating nutrients, runners of plants injected with BA were much stronger. The competition for nutrients among these strong runners made them have a shorter average length compared to runners under other treatments. As a benefit, growers have more options to get healthy and strong runners, while also saving the nutrients and energy to support the mother plant.

The ETH is quite different from the other two PGRs mentioned above, as it eventually decomposes into ethylene, so temperature (high temperatures accelerate ETH movement inside the plant), pH, period of usage, and the plant growth stage can easily influence the effect of ETH [35]. In plant production settings, ETH is applied by spraying, dipping, smearing, or air fumigation, among which spraying is the most commonly used. The ETH can be absorbed by leaves, stems, fruits, and other organs but is mainly taken up through leaf surface absorption. When ETH enters the vascular bundle, it is transported to other tissues and organs as the organic matter moves, so ETH has some mobility within the plant [36]. The ETH that enters the cell is broken down gradually to release ethylene, and then produces its effect on the plant. The ETH is similar to ethylene in that it enhances the synthesis of RNA in cells and promotes the synthesis of proteins. It has also been shown by studies that plants treated with ETH had high protein contents (Figure 2A).

Both crown injection and medium drench of ETH induced dead leaves, albeit to different degrees, and inhibited formation of new leaves and runner in strawberries. Foliar ETH spray did not induce any leaf deaths. The other two PGRs did not have the same leaf-killing and development-inhibiting effects. These results indicate that ETH has differing effects that depend on the application site, or that different plant tissues or organs have different tolerances to ETH, as they do to certain natural phytohormones. Another possible explanation for the differing effects of ETH by application method is the concentration of the solution. In this study, a concentration at 100 mg·L$^{-1}$ was used. The higher the applied concentration, the more likely the occurrence of phytotoxicity. Phytohormones are characterized by low doses but high efficiencies. Some of them, such as IAA, have effects that depend on the concentration, where they promote plant growth at a low concentration but inhibit growth at a high concentration. As an analogue of plant hormone, ETH may have similar characteristics. To clarify the reason and mechanism by which dead leaves were induced by crown injection and medium drench of ETH in this experiment, further research is needed.

Stressful environments can cause the accumulation of reactive oxygen species (ROS) or free radicals in plants. Harsh environments, normal oxygen metabolism, certain chemical reactions, or toxic agents in the environment could force plants to produce such substances that continuously threaten the cells and tissues. The ROS and free radicals are able to disrupt the metabolic activity and cell structure. When this occurs, additional free radicals are produced in a chain reaction that leads to more extensive damage to plants, particularly the oxidation of DNA, proteins, and membrane lipids. Fortunately, plants can defend themselves against such damages via synthesizing antioxidant enzymes such as SOD, POD, and CAT to eliminate stresses [37]. It is true that the use of PGRs could push plants into stressed states. One example is the dead leaves and a low number of newly-grown runners in plants treated with ETH as discussed above. The experimental results also confirmed that the treatments did cause some biological stresses on the strawberry plants, because the activities of antioxidant enzymes in most

of the treated plants were higher than that in the control group. Generally, ETH treatments resulted in higher activities of antioxidant enzymes, and BA treatments, especially by injection, resulted in lower activities of antioxidant enzymes in strawberry. To some extent, BA injection had little toxic and side effects on strawberry mother plants.

Although the PGRs selected were reported to have low toxicity, according to a research in mice, ETH could be harmful for the kidney and liver even in small doses [38]. The CCC was also reported to be toxic on the fertility of mammals such as pigs and mice [39]. The BA toxicity is seldom reported, and thus can be considered as no concern for human and animal safety. For plant safety, improper application or excessively high concentrations of CCC result in severe marginal leaf chlorosis or chlorotic spotting [40].

Typical application methods for PGRs are foliar spray or medium drench, but substrate spray, bulb spray, seed soak, cutting, and liner dips are also used. Each method has advantages and disadvantages, so appropriate methods should be chosen for a particular situation [41]. The injection method used in the experiment was actually similar with cutting and liner dips in that it damages the surface of plant tissues and lets the PGRs into the plant body. It is not easy to inject PGR solutions into strawberry crowns because strawberry do not have blood vessels like human or animals. If we obey the definition of 'injection' strictly, namely, it involves pricking the surface tissues of plants with an injector and then sending the solution inside, as tested, one person can finish one or two plants per minute, and this was the method tried prior to this experiment.

In the second turn of the experiment, a new method for raising efficiency was found and used. An injector was used to make small pores on the crowns, pushed in to let its contents out, without the need for pricking the entire needle inside. This allowed solutions to enter the plant through small pores, and the effectiveness was not compromised at all, compared with the first method in prior experiment discussed above. This allowed for three-four additional plants to be treated per minute. This means that, for example, if a grower wished to control the number and length of 5000 strawberry plants, it would take 1000 man/min (16.7 man/h) under ideal conditions, making this improved injection method practically applicable to real production environments.

*4.2. Conclusions*

The three PGRs had different effects on strawberry runners and mother plants, and the results are variable as affected by their application methods. The most successful combination of PGR and application method in this study was BA injection into crowns, because it achieved expected experimental goal of shortening the average length of runners without harming the mother plants and daughter plants, while increasing the number of new leaves and runners, and the diameter of runners. It is important to note that BA has low toxicity, thus posing little to no risks to human health. Because the aforementioned injection method is easy and efficient, it is possible to lower the cost and time in production setting. Thus, BA injection into crowns is recommended for growers in need of controlling the number and length of runners. As for the other two PGRs, ETH treatment caused many dead leaves and weak runners in mother plants, and it had little effect in controlling runners; CCC solution with injection could shorten runner length, but the effect was less dramatic than BA injection. The other two methods were not observed having any special effects on either cultivar. Runner plants (next generation) grown for 30 days were not affected by any treatments, and they were in similar conditions. The effects of the three PGRs in different concentrations have not been tested yet. A 100 mg·L$^{-1}$ is not necessarily the optimal concentration for the PGRs, and the PGR solutions with different concentrations should also lead to different results. The amount of PGR residue in the fruits has not been tested either, and may be the focus of future experiments.

**Author Contributions:** Conceptualization, B.R.J.; Methodology, B.R.J. and C.L.; Formal Analysis, C.L., Z.G., and H.W.; Resources, B.R.J.; Data Curation, Y.G.P.; Writing–Original Draft Preparation, C.L.; Writing–Review and Editing, B.R.J. and Y.G.P.; Project Administration, B.R.J.; Funding Acquisition, B.R.J., C.L., Z.G., Y.G.P., and H.W.

**Funding:** This research was funded by the Agrobio-Industry Technology Development Program; Ministry of Food, Agriculture, Forestry, and Fisheries; Republic of Korea (Project No. 315004-5). C.L., Z.G., and H.W. were supported by a scholarship from the BK21 Plus Program, Ministry of Education, Republic of Korea.

**Acknowledgments:** This research was supported by the Agrobio-Industry Technology Development Program; Ministry of Food, Agriculture, Forestry, and Fisheries; Republic of Korea (Project No. 315004-5). Chen Liu, Ziwei Guo, and Hao Wei were supported by a scholarship from the BK21 Plus Program, Ministry of Education, Republic of Korea.

**Conflicts of Interest:** The authors declare no conflict of interest.

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
