# Peer review of "PGR and Its Application Method Affect Number and Length of Runners Produced in ‘Maehyang’ and ‘Sulhyang’ Strawberries"

_agronomy, doi:10.3390/agronomy9020059_

Reviewer 1 Report

In the present work, the authors investigate the involvement of PGR on the control the number and length of strawberry (‘Maehyang’ and ‘Sulhyang’) runners. For this purpose the authors had various approaches: Chlormequat chloride (CCC), 6-benzylaminopurine (BA), and ethephon (ETH) at a concentration of 100 mg·L-1 were selected with 3 application ways of injecting in crowns, drenching in roots and spraying on leaves. They conclude that (1) injection was an effective way of giving PGR solution, (2) all 3 PGR solutions with injection to both cultivars made the runners shorter than those in the control group, (3) among 3 PGRs, BA solution behaved the best.
Although the manuscript displays some interesting data concerning the effects of CCC, BA and ethylene on  on the control the number and length of strawberry (‘Maehyang’ and ‘Sulhyang’) runners, it does not extend our knowledge on the mechanism involved in the regulation of growth parameters by CCC and ethylene. The authors limit themselves to study the effects of 3 PGRs only at 100 mg/L. Why the authors did not check other concentrations?
The  experiments seem well executed and this is well documented. However, miss a statement on how many times these experiments were independently repeated. It is of eminent importance to repeat such experiments at least once, but preferably twice.
While not entirely novel, the work has its merits.
TITLE
The paper title is well stated, it is informative and concise.
ABSTRACT
Abstract needs a little improvement. The abstract do not characterize the contents of the paper sufficiently (missing background and aims of work).
MATERIAL AND METHODS
Material and research methods are not presented appropriately and clearly. The description of experiments is very long, not always coherent and in parts difficult to follow.
The authors limit themselves to study the effects of 3 PGRs only at 100 mg/L. Why the authors did not check other concentrations?
RESULTS
The results obtained in this study are interesting. No standard deviation in the results (Tabs).
DISCUSSION
In general, the discussion of results is correct and sufficient.
LITERATURE
The items of literature included in the paper are rather sufficient and adequate to the subject of the paper.

Author Response

Point 1: In the present work, the authors investigate the involvement of PGR on the control the number and length of strawberry (‘Maehyang’ and ‘Sulhyang’) runners. For this purpose the authors had various approaches: Chlormequat chloride (CCC), 6-benzylaminopurine (BA), and ethephon (ETH) at a concentration of 100 mg·L-1 were selected with 3 application ways of injecting in crowns, drenching in roots and spraying on leaves. They conclude that (1) injection was an effective way of giving PGR solution, (2) all 3 PGR solutions with injection to both cultivars made the runners shorter than those in the control group, (3) among 3 PGRs, BA solution behaved the best.

 Although the manuscript displays some interesting data concerning the effects of CCC, BA and ethylene on the control the number and length of strawberry (‘Maehyang’ and ‘Sulhyang’) runners, it does not extend our knowledge on the mechanism involved in the regulation of growth parameters by CCC and ethylene. The authors limit themselves to study the effects of 3 PGRs only at 100 mg/L. Why the authors did not check other concentrations?

Response 1: As the results shown in the manuscript, CCC and ethephon had only limited help to the objectives of this experiment, so there was not much to discuss about the roles of these two PGRs in strawberry. Nevertheless, we still discussed the mobility of CCC and ethephon in strawberry plants and the impacts on plant nutrition and health. Three PGRs combined with three application methods were used in this study, and each treatment had three replicates. This made the whole study to need a lot of plants and experimental space. It is not easy to management and operate with such large number of plants in one experiment. In order to ensure good conditions of plants and experimental results, no other concentrations could be tested in this experiment unfortunately. Nevertheless, the concentration tested was chosen according to the practical production, because there has been no report on testing these PGRs on strawberry plants and information about the proper concentration in strawberry was rarely found, just as was mentioned in the manuscript (Materials and methods 2.2, line141-154). It does not rule out though that we will test other concentrations in our future studies.

Point 2: The experiments seem well executed and this is well documented. However, miss a statement on how many times these experiments were independently repeated. It is of eminent importance to repeat such experiments at least once, but preferably twice.

Response 2: The experiment was repeated once and this was mentioned in the last part of the article (line 434-436). Although ‘injection’ was performed differently in both trials, the results were highly similar.

While not entirely novel, the work has its merits.
TITLE
The paper title is well stated, it is informative and concise.
Thanks for you kindly comment.

ABSTRACT
Point 3: Abstract needs a little improvement. The abstract do not characterize the contents of the paper sufficiently (missing background and aims of work).
Response 3: Abstract has been modified, and background and aims of expt. have been highlighted

MATERIAL AND METHODS
Point 4: Material and research methods are not presented appropriately and clearly. The description of experiments is very long, not always coherent and in parts difficult to follow.

Response 4: ‘Materials and methods’ part was improved, and the superfluous parts were deleted and the means of expression were polished.

Point 5: The authors limit themselves to study the effects of 3 PGRs only at 100 mg/L. Why the authors did not check other concentrations?

 Response 5: Three PGRs combined with three application methods were used in this study, and each treatment had three replicates. This made the whole study to need a lot of plants and experimental space. It is not easy to management and operate with such large number of plants in one experiment. In order to ensure good conditions of plants and experimental results, no other concentrations could be tested in this experiment unfortunately. Nevertheless, the concentration tested was chosen according to the practical production, because there has been no report on testing these PGRs on strawberry plants and information about the proper concentration in strawberry was rarely found, just as was mentioned in the manuscript (Materials and methods 2.2, line141-154). It does not rule out though that we will test other concentrations in our future studies.
RESULTS
Point 6: The results obtained in this study are interesting. No standard deviation in the results (Tabs).
Response 6: Standard deviations have been added in tables.

DISCUSSION
In general, the discussion of results is correct and sufficient.
Thanks for you kindly comment.

LITERATURE
The items of literature included in the paper are rather sufficient and adequate to the subject of the paper.

Thanks for you kindly comment.

Reviewer 2 Report

The language needs significant improvement throughout the whole document. I recommend to hire a professional translation service.

The hypothesis and the objectives of the study need some clarification and need to be clearly mentioned in the introduction. 

Material and Methods need clarification

Greenhouse conditions (temperature/photoperiod) are not specified. Those factors have a major impact on runner/flower production in strawberry.

Figure legends need better description of what is seen in the figures

It is not clear how some of the assessed parameters (protein content/photosystem activity etc.) are practically related to the described process of propagating strawberry plants in fruiting fields.

The grower recommendation to inject BA in planting holes, plus the provided basic labor analysis is not substantiated by the results or anywhere else in the document. 

The practice of propagation during the fruit production/harvest season is not a good practice, for reasons such as the potential spread of disease, and of course for the substantial loss of fruit production. This is true even for perpetual flowering strawberry types, which have the potential to produce flowers and runners all season long. In US, Europe, China and elsewhere, strawberry plants are usually propagated in nurseries for several generations, and the last generation of plants is then conditioned to produce flowers rather than runners.

However, even in conditioned plants, and especially in perpetual flowering type strawberry plants, rates of runner and flowering are controlled by temperature, photoperiod and nutrition. Those factors are relatively easy to manipulate in a greenhouse setting. 

The practice of propagation during fruit production might locally be important, but certainly is not a global practice. 

While the use of hormones and growth regulators in vegetative propagation of strawberry needs to be researched, and is valuable information for nurseries, I can't recommend this manuscript for publication in the form it was presented. Above mentioned major flaws in language, structure and  missing of important information in the experimental setup (temperature/light?, labor hours?) are the reason..  

Author Response

Point 1: The language needs significant improvement throughout the whole document. I recommend to hire a professional translation service.

Response 1: Thanks for your advice. The language has been revised by a professional editor as you suggested.

Point 2: The hypothesis and the objectives of the study need some clarification and need to be clearly mentioned in the introduction.

Response 2: Hypothesis and the objectives in the introduction were clearly mentioned and highlighted in the introduction. The hypothesis of this experiment comes from our previous studies. It is found that some of PGRs can control the length of plant organs, such as potato runners. Therefore, we believed that PGRs also could be useful to control strawberry runners. Different PGRs have different effects, the method of application may also change the effect of PGR. Thus, the objectives were to investigate effects of CCC, BA and ETH on strawberry (runners, mother plants, and runner plants) and also to find out appropriate PGRs and their application methods to control number and length of the runners.

Point 3: Material and Methods need clarification

Greenhouse conditions (temperature/photoperiod) are not specified. Those factors have a major impact on runner/flower production in strawberry.

Response 3: The glasshouse conditions have been already mentioned in the manuscript (from lines 119 to 121, and lines 133 to 134).

Point 4: Figure legends need better description of what is seen in the figures

Response 4: Figure legends were revised by a professional editor.

Point 5: It is not clear how some of the assessed parameters (protein content/photosystem activity etc.) are practically related to the described process of propagating strawberry plants in fruiting fields.

Response 5: Protein content can be used to calculate the activities of antioxidant enzymes; on the other hand, some PGRs, such as ETH, can promote protein synthesis (from lines 389 to 391). By comparing contents of protein, soluble sugar and starch in plants treated with PGR with those of plants in the control, effects of PGRs and their application methods on these materials can be known, and these assessed parameters are crucial for the growth and reproduction of plants. It can be realized from the data that different PGRs and application methods can cause differences in the content of those materials (Fig. 2). In the discussion part, previous relevant studies and our speculation were also mentioned. In addition, photosystem activity was not mentioned in the article, but the activities of antioxidant enzymes (Fig. 3). The activities of antioxidant enzymes can be used to understand the stress status of plants, so as to infer whether PGRs and their application methods have caused stress on strawberry mother plants.

Point 6: The grower recommendation to inject BA in planting holes, plus the provided basic labor analysis is not substantiated by the results or anywhere else in the document.

Response 6: In this experiment a BA solution was injected into strawberry crowns, not in planting holes. It is true that basic labor analysis was not substantiated by the results, because this issue was basically not an objective of the experiment. The objective was to understand the specific effects of PGRs on strawberry, rather than to promote growers to apply this technique yet. Labor hours were calculated based on the time spent during the treatment, such as how many strawberry plants each person could be injected per one minute (From lines 440 to 441). It was only a rough estimate to show that whether the injection of PGR solution can be used in real production operations.

Point 7: The practice of propagation during the fruit production/harvest season is not a good practice, for reasons such as the potential spread of disease, and of course for the substantial loss of fruit production. This is true even for perpetual flowering strawberry types, which have the potential to produce flowers and runners all season long. In US, Europe, China and elsewhere, strawberry plants are usually propagated in nurseries for several generations, and the last generation of plants is then conditioned to produce flowers rather than runners.

Response 7: In Republic of Korea, especially in the southern areas of the country, strawberry ‘Maehyang’ and ‘Sulhyang’ are grown in large quantities in greenhouses. These two cultivars are not perpetual flowering strawberry types. Strawberry growers usually control environment conditions to force plants to bloom as early as possible in the cultivation season, usually by late October to early November. From November to following February is the season to produce/harvest fruits. March to April is a suitable time for runners growing, while May and June are the best time to harvest runner plants to produce new mother plants. Runner plants are often collected, sold or planted to expand the planting area. Then July to September is the time for the new strawberry generations to grow. Yes, you are right on the plant quality issue. Such production pattern is practiced in Republic of Korea, year after year. The growth and maintenance of strawberry plants in our experiment was in line with the local production practices, and the expt. time (May-June) was not during the fruit production/harvest season.

Point 8: However, even in conditioned plants, and especially in perpetual flowering type strawberry plants, rates of runner and flowering are controlled by temperature, photoperiod and nutrition. Those factors are relatively easy to manipulate in a greenhouse setting. 

Response 8: Yes, you are right and thank you for the useful information on strawberry physiology. This study did not aim to control the growth rate of runners, nor to control flowering period as one of the objectives. Instead, we were interested in the possibility of controlling runner length and number. Although manual control of environmental factors, such as temperature, photoperiod and nutrition, may achieve similar goals, the use of PGRs is another way and may not be so difficult to implement.

Point 9: The practice of propagation during fruit production might locally be important, but certainly is not a global practice. 

Response 9: Our study was conducted for application on only Korean local cultivars and cultivation practices. However, if our results can be known by more strawberry researchers, more people may try other global cultivars as well as more PGRs and application methods to contribute to the strawberry industry together.

Point 10: While the use of hormones and growth regulators in vegetative propagation of strawberry needs to be researched, and is valuable information for nurseries, I can't recommend this manuscript for publication in the form it was presented. Above mentioned major flaws in language, structure and missing of important information in the experimental setup (temperature/light? labor hours?) are the reasons.

Response 10: Thanks for your kindly comments. Authors have added the information you suggested to be missing in Materials and Methods and also in Discussion sections of the manuscript.

Round  2

Reviewer 2 Report

Improvement made in language, and the clarification of hypothesis and objectives.

However, sentences such as those in the Abstract (15-17):'However, overlong runners would take away a high percentage of total nutrients and energy from the mother plant, which may lead to poor growth or reduced output.' are still wrong and would need to be rewritten.

Runners can't 'take away' any nutrients or even energy. Nutrients are 'translocated', which is a completely different process than taking something away. 

Plants rarely 'take something away'. While it is clear what the author means, it is described not correctly. A further improvement of language by a native speaker would definitely improve the manuscript. However, objectives, procedures and results are clearly described.

Author Response

Point: Improvement made in language, and the clarification of hypothesis and objectives. However, sentences such as those in the Abstract (15-17): 'However, overlong runners would take away a high percentage of total nutrients and energy from the mother plant, which may lead to poor growth or reduced output' are still wrong and would need to be rewritten.

Runners can't 'take away' any nutrients or even energy. Nutrients are 'translocated', which is a completely different process than taking something away.

Plants rarely 'take something away'. While it is clear what the author means, it is described not correctly. A further improvement of language by a native speaker would definitely improve the manuscript. However, objectives, procedures, and results are clearly described.

Response: Thanks a lot for your constructive suggestions. The mistakes you mentioned have been already corrected in the manuscript (from lines 15 to 17, and lines 50 to 52).
